# QH9: A Quantum Hamiltonian Prediction Benchmark for QM9 Molecules

**Haiyang Yu**[*]
Texas A&M University
College Station, TX 77843
haiyang@tamu.edu

**Meng Liu**[*]
Texas A&M University
College Station, TX 77843
mengliu@tamu.edu

**Youzhi Luo**
Texas A&M University
College Station, TX 77843
yzluo@tamu.edu

**Alex Strasser**
Texas A&M University
College Station, TX 77843
alexstrasser16410@tamu.edu

**Xiaofeng Qian**[†]
Texas A&M University
College Station, TX 77843
feng@tamu.edu

**Xiaoning Qian**[†]
Texas A&M University
College Station, TX 77843
xqian@ece.tamu.edu

**Shuiwang Ji**[†]
Texas A&M University
College Station, TX 77843
sji@tamu.edu

## Abstract

Supervised machine learning approaches have been increasingly used in accelerating electronic structure prediction as surrogates of first-principle computational methods, such as density functional theory (DFT). While numerous quantum chemistry datasets focus on chemical properties and atomic forces, the ability to achieve accurate and efficient prediction of the Hamiltonian matrix is highly desired, as it is the most important and fundamental physical quantity that determines the quantum states of physical systems and chemical properties. In this work, we generate a new Quantum Hamiltonian dataset, named as QH9, to provide precise Hamiltonian matrices for 999 molecular dynamics trajectories and 130,831 stable molecular geometries, based on the QM9 dataset. By designing benchmark tasks with various molecules, we show that current machine learning models have the capacity to predict Hamiltonian matrices for arbitrary molecules. Both the QH9 dataset and the baseline models are provided to the community through an open-source benchmark, which can be highly valuable for developing machine learning methods and accelerating molecular and materials design for scientific and technological applications. Our benchmark is publicly available at https://github.com/divelab/AIRS/tree/main/OpenDFT/QHBench.

## 1 Introduction

Machine learning methods have shown great potential in accelerating computations in quantum chemistry tasks [Zhang et al., 2023, Huang et al., 2023, Kirkpatrick et al., 2021]. For example, a variety of invariant geometric deep learning methods have been developed to encode pairwise distances and bond angles in molecular and materials systems [Schütt et al., 2018, Gasteiger et al., 2020, 2021, Liu et al., 2022, Wang et al., 2022, Lin et al., 2023, Yan et al., 2022, Liao et al., 2023,

---

[*]Equal contribution
[†]Equal senior contribution

37th Conference on Neural Information Processing Systems (NeurIPS 2023) Track on Datasets and Benchmarks.

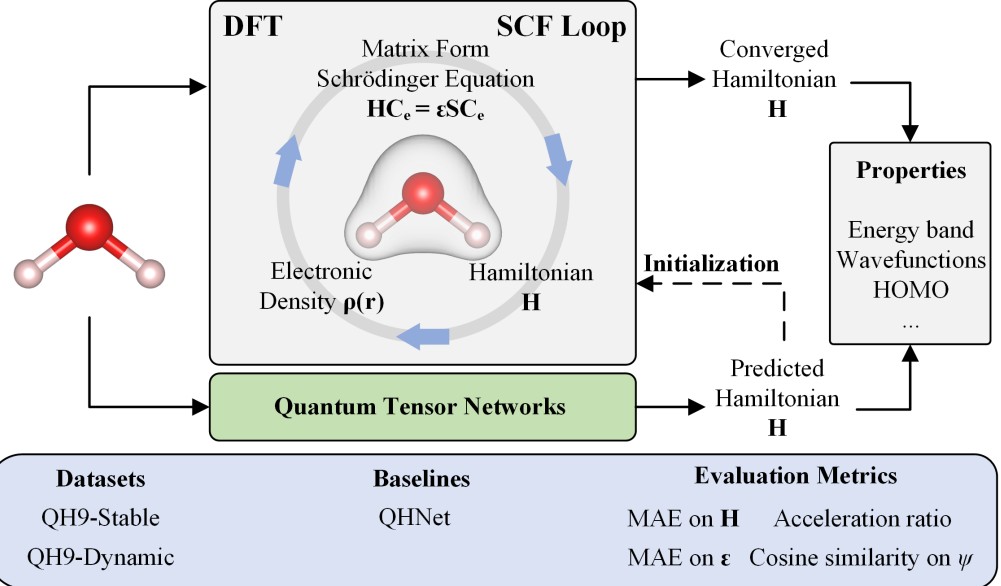

Figure 1: The target and content of the propose QH9 dataset and benchmark. Quantum tensor networks are built for predicting the target Hamiltonian matrix, facilitating the optimization loop within DFT by providing a precise approximation. Within this benchmark, stable and dynamic datasets are generated for training powerful quantum tensor networks and comprehensive evaluation metrics are proposed to measure the prediction quality.

Schütt et al., 2021, Batzner et al., 2022, Liu et al., 2021] to accelerate the prediction of their chemical properties as data-driven surrogate approximations. To enhance the prediction of vectorial properties, such as force fields, equivariant deep learning methods have been developed to capture permutation, translation, and rotation equivariance for equivariant property prediction [Satorras et al., 2021, Schütt et al., 2021, Thölke and Fabritiis, 2022, Thomas et al., 2018, Batzner et al., 2022, Fuchs et al., 2020, Liao and Smidt, 2023, Anderson et al., 2019, Brandstetter et al., 2022, Batatia et al., 2022]. To support and facilitate the development of machine learning methods on quantum chemistry property prediction, many datasets have been generated to benchmark the respective tasks on molecular property prediction [Blum and Reymond, 2009, Ruddigkeit et al., 2012, Ramakrishnan et al., 2014, Wang et al., 2009, Nakata and Shimazaki, 2017], catalyst prediction [Chanussot et al., 2021, Tran et al., 2023], and force field prediction [Chmiela et al., 2017, 2023].

In addition to these quantum chemistry prediction tasks, the quantum Hamiltonian is another significant and fundamental physical property that determines the quantum states and various materials properties [Marzari and Vanderbilt, 1997, Souza et al., 2001, Qian et al., 2010, Marzari et al., 2012, Bai et al., 2022]. The quantum Hamiltonian can be calculated using Density Functional Theory (DFT) [Hohenberg and Kohn, 1964, Kohn and Sham, 1965] with a time complexity of $O(n^3 T)$, where $n$ represents the number of electrons and $T$ denotes the number of optimization steps required to achieve convergence. Given the high computational complexity of the DFT algorithms, accelerating such calculations for novel molecular and materials systems becomes a desirable but challenging task. To tackle this challenge, machine learning methods, such as quantum tensor networks [Li et al., 2022, Gong et al., 2023, Schütt et al., 2019, Yu et al., 2023, Unke et al., 2021], provide a highly promising approach for accelerating the DFT algorithms. These networks directly predict the final Hamiltonian matrix given the input 3D geometries, resulting in significant acceleration of calculations by orders of magnitude.

Unlike invariant chemical properties, Hamiltonian matrices obey intrinsic block-by-block matrix equivariance. This equivariance can be represented by the rotation Wigner D-Matrix, which may contain higher order rotations beyond 3D space. In order to make physically meaningful predictions, it is important to design quantum tensor network architectures that preserve this equivariance property. To perform systematic and in-depth study of this new task, there is a clear need to generate large-scale quantum tensor datasets and benchmarks. Currently, existing quantum Hamiltonian datasets

include the MD17 [Schütt et al., 2019, Gastegger et al., 2020] and mixed MD17 [Yu et al., 2023] datasets, which consist of data for a single and four molecules, respectively. Recent public dataset NablaDFT [Khrabrov et al., 2022] contains Hamiltonian matrices for molecular conformations.

To provide a realistic dataset and comprehensive evaluation, we generate a new quantum tensor dataset named QH9. This dataset contains Hamiltonian matrices for 130,831 stable molecular geometries and 999 molecular dynamic trajectories. In order to provide comprehensive studies for quantum tensor networks, we have designed four specific tasks. The first two tasks, QH9-stable-id and QH9-stable-ood, aim to explore the performance of the networks in both in-distribution and out-of-distribution scenarios, specifically focusing on stable molecular geometries. The QH9-dynamic-geo task follows the setting of the mixed MD17, containing the same molecule with different geometries in the training, validation, and test. On the other hand, the QH9-dynamic-mol task splits the trajectories based on different molecules. Finally, we evaluate the transferability of the trained models on molecules with larger sizes, thereby testing the models' ability to generalize beyond the training dataset. To demonstrate the quality of the predicted Hamilton matrix, we use four metrics. These metrics are based on the Mean Absolute Error (MAE) of the predicted Hamiltonian matrix $\boldsymbol{H}$, as well as the derived properties such as orbital energies $\boldsymbol{\epsilon}$ and electronic wavefunction $\psi$. Furthermore, to evaluate the quality of the predicted Hamiltonian in accelerating DFT calculations, we calculate the DFT optimization ratio by taking the model predictions as DFT initialization. The target and content of the proposed QH9 dataset and benchmark are demonstrated in Figure 1.

## 2 Background and Related Works

### 2.1 Density Functional Theory (DFT)

Modeling the quantum states of physical systems is a central topic in computational quantum physics and chemistry. It aims to solve the Schrödinger equation [Schrödinger, 1926], which describes the electronic states shown as

$$\hat{H}\Psi\left(\boldsymbol{r}_1, \cdots, \boldsymbol{r}_n\right) = E\Psi\left(\boldsymbol{r}_1, \cdots, \boldsymbol{r}_n\right), \tag{1}$$

where $\Psi\left(\boldsymbol{r}_1, \cdots, \boldsymbol{r}_n\right)$ is the $n$-electronic wavefunctions and $\boldsymbol{r}$ is the 3D coordinates. Electronic eigenvalues and wavefunctions play an important role in calculating numerous crucial physical properties, including the Highest Occupied Molecular Orbital (HOMO), the Lowest Unoccupied Molecular Orbital (LUMO), charge density and many others. However, due to the exponentially expanding input Hilbert space with the number of electrons, the computational cost to directly calculate many-electronic wavefunctions is extremely high. Therefore, various methods are proposed to approximate the solutions, such as the Hartree-Fock (HF) method [Szabo and Ostlund, 2012] that approximates the wavefunction itself, or density functional theory [Hohenberg and Kohn, 1964] that approximates the electron density. While the HF method scales with the number of electrons $n$ as $O(n^4 T)$, DFT scales with $O(n^3 T)$ and therefore DFT is better suited for large-scale systems. DFT is based on the key discovery that the total energy and thus all ground-state properties of a system are uniquely determined by the ground-state electron density [Kohn and Sham, 1965].

Both of these approaches divide an $n$-electron system into a set of $n$ non-interacting one-electron wavefunctions $\psi_i(\boldsymbol{r_i})$, also called molecular orbitals in molecular systems. These one-electron orbitals can then be approximated by a linear combination of basis functions $\phi_j(\boldsymbol{r})$ as $\psi_i(\boldsymbol{r}) = \sum_j C_{ij}\phi_j(\boldsymbol{r})$. The basis functions can be represented in analytical forms, such as Slater-type orbitals (STOs), Gaussian-type orbitals (GTOs), or plane waves, under numerical approximations for obtaining the coefficients matrix $\boldsymbol{C}$. With these approximations, the original Schrödinger Equation (1) for electrons can be transformed into a matrix form as

$$\boldsymbol{H}\boldsymbol{C}_i = \boldsymbol{\epsilon}_i \boldsymbol{S}\boldsymbol{C}_i, \tag{2}$$

where $\boldsymbol{H}$ is the Hamiltonian matrix, $\boldsymbol{S}$ is the overlap matrix, and $\boldsymbol{\epsilon}_i$ is the energy for the $i$-th orbital. The Hamiltonian matrix can be decomposed into the sum

$$\boldsymbol{H} = \boldsymbol{H}_{eN} + \boldsymbol{H}_{ee} + \boldsymbol{H}_{XC}, \tag{3}$$

which describes electron-nucleus interactions ($\boldsymbol{H}_{eN}$), electron-electron interactions ($\boldsymbol{H}_{ee}$, including kinetic energy and electron–electron Coulomb repulsion energy), and exchange-correlation energy ($\boldsymbol{H}_{XC}$). These matrices take the electron density $\boldsymbol{\rho}(\boldsymbol{r})$ as an input to evaluate the Hamiltonian matrix.

The exchange-correlation energy functional used in this paper was B3LYP [Lee et al., 1988, Becke, 1993], which is a hybrid functional that includes both the exchange energy from the HF method as well as a correlation potential. Thus, the complexity of using B3LYP in DFT is $O(n^4T)$, which is the same as HF. We implement the GTO basis set Def2SVP [Weigend and Ahlrichs, 2005] that incorporates aspects of DFT, namely, an exchange-correlation potential, in order to more accurately capture electron-electron interactions compared to the HF method, which uses a mean field approximation of electron density.

Equation (2) is satisfied for the final Hamiltonian matrix and its coefficient matrix once self-consistency is achieved using direct inversion in the iterative subspace (DIIS) [Pulay, 1980, 1982]. The equation is solved iteratively by building and solving the Hamiltonian and coefficient matrices, constructing an error vector based on a linear combination of energy differences in the previous steps, then diagonalizing and recalculating $\boldsymbol{H}$ until the error vector is below a convergence threshold. In our QH9 datasets, we provide the Hamiltonian matrix $\boldsymbol{H}$ used to train quantum tensor networks for directly predicting the Hamiltonian matrix.

## 2.2 Group Equivariance and Equivariant Matrices

In many quantum chemistry problems, the molecular property to be predicted (e.g., energy and force) is internally invariant or equivariant to transformations in SE(3) group, including rotations and translations. Formally, for an $n$-atom molecule whose 3D atom coordinates are $\boldsymbol{r}_1, ..., \boldsymbol{r}_n$, any transformation in SE(3) group can be described as changing the 3D atom coordinates to $\boldsymbol{R}\boldsymbol{r}_1 + \boldsymbol{t}, ..., \boldsymbol{R}\boldsymbol{r}_n + \boldsymbol{t}$. Here, the translation vector $\boldsymbol{t} \in \mathbb{R}^3$ is an arbitrary 3D vector, and the rotation matrix $\boldsymbol{R} \in \mathbb{R}^{3\times 3}$ satisfies that $\boldsymbol{R}^T\boldsymbol{R} = \boldsymbol{I}, |\boldsymbol{R}| = 1$. Let $\boldsymbol{f}(\cdot)$ map the 3D atom coordinates to an $(2\ell + 1)$-dimensional prediction target vector, we say $\boldsymbol{f}$ is order-$\ell$ SE(3)-equivariant if

$$\boldsymbol{f}(\boldsymbol{R}\boldsymbol{r}_1 + \boldsymbol{t}, ..., \boldsymbol{R}\boldsymbol{r}_n + \boldsymbol{t}) = D^\ell(\boldsymbol{R})\boldsymbol{f}(\boldsymbol{r}_1, ..., \boldsymbol{r}_n) \tag{4}$$

holds for any rotation matrix $\boldsymbol{R}$ and translation vector $\boldsymbol{t}$, where $D^\ell(\boldsymbol{R}) \in \mathbb{C}^{(2\ell+1)\times(2\ell+1)}$ is the order-$\ell$ Wigner-D matrix of $\boldsymbol{R}$ (please refer to Section A.3 of Brandstetter et al. [2022] and Section A.2.1 of Poulenard et al. [2022] for more background information about the Wigner-D matrix). To accurately predict SE(3)-equivariant properties, an effective approach is to develop neural network models that are designed to maintain the same equivariance relations between inputs and outputs as in Equation (4). Recently, many studies have proposed SE(3)-equivariant neural network architectures by using SE(3)-invariant feature encoding [Schütt et al., 2018, Gasteiger et al., 2020, 2021, Liu et al., 2022], tensor product operations [Thomas et al., 2018, Brandstetter et al., 2022, Liao and Smidt, 2023], or atomic cluster expansion framework [Batatia et al., 2022, Drautz, 2019, Dusson et al., 2022, Kovács et al., 2021, Musaelian et al., 2023].

Different from vector-like molecular properties, the Hamiltonian matrix $\boldsymbol{H}$ has a much more complicated SE(3) equivariance pattern that is associated with the intrinsic angular momentum of the atomic orbital pairs. In computational quantum chemistry algorithms such as DFT, the Hamiltonian matrix $\boldsymbol{H}$ obtained from DFT calculations can be used to represent the interactions between these atomic orbitals, and the block $\boldsymbol{H}_{ij}$ in Hamiltonian matrix represents the interactions between the atomic orbitals $i$ in atom $a_i$ with angular quantum number $\ell_i$ and atomic orbitals $j$ in atom $a_j$ with angular quantum number $\ell_j$, and the shape of this block $\boldsymbol{H}_{ij}$ is $(2\ell_i + 1) \times (2\ell_j + 1)$. Usually, the atomic orbitals are arranged sequentially for the orbitals in the same atom and with the same angular quantum number. For example, $\boldsymbol{H}_{ij}$ can be located within the $s_i$-th to $(s_i + 2\ell_i)$-th row, and the $s_j$-th to $(s_j + 2\ell_j)$-th column of Hamiltonian matrix $\boldsymbol{H}$. Specifically, its SE(3) equivariance can be described as

$$\boldsymbol{H}_{ij}\left(\boldsymbol{\rho}(\boldsymbol{R}\boldsymbol{r} + \boldsymbol{t})\right) = D^{\ell_i}(\boldsymbol{R})\boldsymbol{H}_{ij}\left(\boldsymbol{\rho}(\boldsymbol{r})\right)D^{\ell_j}(\boldsymbol{R}^T), \tag{5}$$

where $\boldsymbol{\rho}(\boldsymbol{r})$ is the electronic density at positioin $\boldsymbol{r}$ and Hamiltonian matrix $\boldsymbol{H}$ is a function of the electronic density $\boldsymbol{\rho}(\boldsymbol{r})$ in the DFT algorithm. In other words, the SE(3) equivariance of different submatrices in $\boldsymbol{H}$ has different mathematical forms, which is much more complicated than the SE(3) equivariance of vector-like molecular properties. Hence, it is much more challenging to develop SE(3)-equivariant neural network architectures for the prediction of Hamiltonian matrices. Nowadays, only a few studies [Li et al., 2022, Gong et al., 2023, Yu et al., 2023, Unke et al., 2021] have made initial exploration in this direction.

### 2.3 Datasets for Quantum Chemistry

To facilitate the usage of machine learning models to predict chemistry properties and accelerate simulations, numerous quantum chemistry datasets have been built to provide extensive and faithful data. Here, we introduce several existing datasets that have been constructed for different tasks respectively, including molecular property prediction, catalyst modeling, molecular force field prediction, and molecular Hamiltonian matrix prediction. For molecular property prediction, the QM7 [Blum and Reymond, 2009] dataset was initially constructed using 7,165 molecules from the organic molecule database GDB-13 [Blum and Reymond, 2009], with each selected molecule having no more than 7 heavy atoms. The primary purpose of creating the QM7 dataset is to provide atomization energies as the target molecular property. Then QM9 [Ramakrishnan et al., 2014, Schütt et al., 2018] was built based on GDB-17 [Ruddigkeit et al., 2012] to provide 134k stable small organic molecules with no more than 9 heavy atoms in each molecule. Moreover, it provides 13 different important quantum chemistry properties, including HOMO and LUMO energies. Based on the molecules from PubQChem [Wang et al., 2009, 2017, Kim et al., 2019, 2021, 2023], PubQChemQC [Nakata and Shimazaki, 2017] provides 3M ground-state molecular structures as well as the HOMO-LUMO gap and excitation energies for 2M molecules. In addition to the molecular property datasets, OC20 [Chanussot et al., 2021] and OC22 [Tran et al., 2023] were developed to provide the data of interactions of catalysts on material surfaces. They provide the geometries of the initial structures to predict the final structures or energies as well as the relaxation trajectories with energy and atomic forces. For the molecular force field prediction datasets, MD17 [Chmiela et al., 2017] and MD22 [Chmiela et al., 2023] contain atomic forces for molecular and supramolecular trajectories respectively as valuable datasets to develop machine learning methods. The last category is the Hamiltonian matrices datasets. MD17 [Schütt et al., 2019, Gastegger et al., 2020] provides the Hamiltonian matrices for single molecular dynamic trajectories to study the Hamiltonian matrices for molecules with various geometries. Building upon this dataset, mixed MD17 [Yu et al., 2023] combines four molecular trajectories in the MD17 to study Hamiltonian matrix prediction tasks with multiple molecules. NablaDFT [Khrabrov et al., 2022] has million of Hamiltonian matrices for molecular conformers and provides the MAE on predicted Hamiltonian matrices with model trained with in-distribution data split. Alongside the increasing interest in Hamiltonian matrix prediction, there is a growing need for datasets that include Hamiltonian matrices with a greater number of molecules and benchmark with comprehensive evaluation to facilitate the subsequent studies.

## 3 Datasets, Tasks, Methods, and Metrics

### 3.1 Datasets

**Dataset Generation.** For the QH9 dataset, we use open-source software PySCF [Sun et al., 2018, 2020] to conduct computational quantum chemistry calculations. In the QH9, there are two sub datasets. The first one is the QH9-stable dataset containing Hamiltonian matrices for 130,831 molecules with their geometries. We obtain the molecules and their geometries in QH9-stable from the QM9 version with 130,831, which is widely used in molecular property prediction tasks in the literature [Schütt et al., 2018, Gasteiger et al., 2020, Liu et al., 2022, Wang et al., 2022]. Note that we cover all the molecules in this QM9 version. The second one is the QH9-dynamic dataset, it has molecular trajectories for 999 molecules and each trajectory contains 100 geometries. To obtain the accurate Hamiltonian matrices for this dataset, we set the hyper-parameters of the DFT algorithms to a tight level. Specifically, we set the grid density level to 3 to calculate accurate electronic density, and the SCF convergence condition is set to SCF tolerance of $10^{-13}$ and gradient threshold of $3.16 \times 10^{-5}$ to ensure the the final states achieve tight convergence. For the density functional, we select the B3LYP exchange-correlation functional to conduct DFT calculations, and GTO orbital basis Def2SVP is selected to approximate the electronic wavefunctions. To accelerate and achieve the convergence of SCF algorithm, we use DIIS algorithm with consideration of the 8 previous steps. For the QH9-dynamic dataset, molecular dynamics simulations are conducted under the microcanonical ensemble, where the number of particles, volume, and energy remain constant (NVE). The temperature is set to 300K, and the time step for recording the molecular trajectory is set to $0.12$ fs with $1,000$ total steps. In the dataset collection, we sampled 100 time steps for each trajectory with a timestep of 1.2 fs. The generated QH9 dataset is available at Zenodo (`https://zenodo.org/records/8274793`) and GitHub (`https://github.com/divelab/AIRS/tree/main/OpenDFT/QHBench/QH9`).

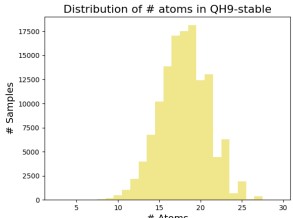
(a) The histogram of the molecule size for QH9-stable dataset.

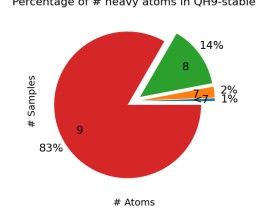
(b) Number of heavy atom percentage in QH9-stable dataset.

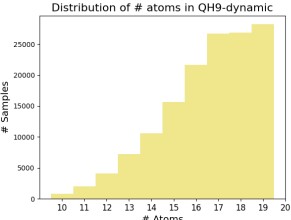
(c) The histogram of the molecule size for QH9-dynamic.

Figure 2: The dataset statistics on QH9-stable and QH9-dynamic, including molecule size distribution and percentage of molecules with different number of heavy atoms.

Table 1: The statistics of our defined four tasks.

| Task | # Total geometries | # Molecules | # Training/validation/testing geometries |
|---|---|---|---|
| QH9-stable-id | $130,831$ | $130,831$ | $104,664/13,083/13,084$ |
| QH9-stable-ood | $130,831$ | $130,831$ | $104,001/17,495/9,335$ |
| QH9-dynamic-geo | $99,900$ | $999$ | $79,920/9,990/9,990$ |
| QH9-dynamic-mol | $99,900$ | $999$ | $79,900/9,900/10,100$ |

**Dataset Statistics.** The statistical data, including the number of molecules and geometries for QH9-stable and QH9-dynamic, is presented in Table 1. These molecules consist of no more than 9 heavy atoms and are composed of four specific heavy atoms: carbon (C), nitrogen (N), oxygen (O), and fluorine (F). The distribution of molecule size for QH9-stable and QH9-dynamic is shown in Figure 2a and Figure 2c. Meanwhile, the percentage of molecules in QH9-stable with different the number of heavy atoms is shown in 2b.

## 3.2 Tasks

To comprehensively evaluate the quantum Hamiltonian prediction performance, we define the following tasks based on the obtained stable and dynamic geometries in the QH9 dataset.

**QH9-stable-id.** We first randomly divide the obtained stable geometries in QH9 into three subsets, including $80\%$ for training, $10\%$ for validation, and $10\%$ for testing. This serves as the basic evaluation task for predicting quantum Hamiltonian matrices.

**QH9-stable-ood.** We further split the stable geometries in QH9 by molecular size based on the number of constituting atoms. The training set consists of molecules with 3 to 20 atoms, maintaining a similar number of training samples as in the QH9-stable-id split. The validation set includes molecules with 21 to 22 atoms, while the testing set has molecules with 23 to 29 atoms. This task allows for an evaluation of the model's generalization ability under an out-of-distribution training setup.

**QH9-dynamic-geo.** For this split and the following QH9-dynamic-mol split, there are 999 molecular dynamics trajectories, while each trajectory includes 100 geometries. In QH9-dynamic-geo, the split is performed geometry-wise. Specifically, for each molecule, 100 geometries are randomly divided into 80 for training, 10 for validation, and 10 for testing. Here, the molecules in the test set are visible during training but the geometric structures are different from training structures.

**QH9-dynamic-mol.** In this QH9-dynamic-mol split, the 999 molecules are divided into training, validation, and testing subsets in a ratio of $0.8/0.1/0.1$. Importantly, different from the above QH9-dynamic-geo setup, all 100 geometries corresponding to a specific molecule are grouped together and assigned to the same subset. This setup introduces a more challenging task than QH9-dynamic-geo since the geometries in the testing set correspond to different molecules as those in training.

## 3.3  Methods

To predict the quantum Hamiltonian matrix, several quantum tensor networks have been proposed [Li et al., 2023, Gong et al., 2023]. SchNOrb [Schütt et al., 2019] uses pairwise distance and direction as the input geometric information to predict the final Hamiltonian matrix. However, SchNOrb lacks the ability to ensure matrix equivariance and relies on data augmentation techniques to encourage equivariance. Another network, DeepH [Li et al., 2022], uses invariant local coordinate systems and a global coordinate system to handle the equivariance challenge. It uses the geometric features within these invariant local coordinate systems to predict invariant Hamiltonian matrix blocks. Next, as a post-processing step, DeepH applies a rotation using Wigner D-Matrix to transform the Hamiltonian matrix blocks from local coordinate systems back to the global coordinate system. Currently, DeepH is applied on predicting Hamiltonian matrices for materials. PhiSNet [Unke et al., 2021] uses an equivariant model architecture that inherently guarantees matrix equivariance. However, current implementation of PhiSNet is limited to supporting single molecule. This limitation arises from the design of the matrix prediction module in PhiSNet, which is designed to predict matrices for the same molecules with fixed matrix size. Therefore, equivariant quantum tensor network QHNet [Yu et al., 2023] is selected as the main baseline method in the QH9 benchmark currently. QHNet has an extendable expansion module that is built upon intermediate full orbital matrices, enabling its capability to effectively handle different molecules. This flexibility allows QHNet to accommodate various molecules in the QH9 benchmark.

## 3.4  Metrics

To evaluate the quality of the predicted Hamiltonian matrix, we adopt several metrics that are used to measure both approximation accuracy and computational efficiency.

**MAE on Hamiltonian matrix H.** This metric calculates the Mean Absolute Error (MAE) between the predicted Hamiltonian matrix and the ground-truth labels from DFT calculation. Each Hamiltonian matrix consists of diagonal blocks and non-diagonal blocks, representing the interactions within individual atoms and the interactions between pairs of atoms, respectively. When the atom pair is distant, the values in the Hamiltonian matrix blocks are typically close to zero. Consequently, as the molecules increase in size, the proportion of distant atom pairs also increases, causing the overall mean value of the Hamiltonian matrix to decrease. Hence, in the subsequent experiments, we compare the MAEs of the diagonal and non-diagonal blocks separately as well as the total MAE on the Hamiltonian matrix.

**MAE on occupied orbital energies $\epsilon$.** Orbital energy, which includes the Highest Occupied Molecular Orbital (HOMO) and Lowest Unoccupied Molecular Orbital (LUMO) energies, is a highly significant chemical property. It can be determined by diagonalizing the Hamiltonian matrix using Equation 2. Hence, this metric can serve as a measure to reflect the quality of the predicted Hamiltonian matrix in accurately deducing the desired property. Specifically, it calculates the MAE on all the occupied molecular orbital energies $\epsilon$ derived from the predicted and the ground-truth Hamiltonian matrix.

**Cosine similarity of orbital coefficients $\psi$.** Electronic wavefunctions can describe the quantum states of molecular systems and are used to derive a range of chemical properties. In order to measure the similarity between the ground-truth wavefunctions and the predicted wavefunctions, we calculate the cosine similarity of the coefficients for the occupied molecular orbitals $\psi$. The corresponding coefficients $\mathbf{C}$ are derived from the predicted and ground-truth Hamiltonian matrix shown in Equation 2.

**Acceleration ratio.** Besides the metrics assessing molecular properties, several acceleration ratios, such as achieved ratio and error-level ratio, are proposed to measure the quality of the predicted Hamiltonian matrix in accelerating DFT calculation. Specifically, the achieved ratio calculates the ratio of the number of optimization steps between initializing with the predicted Hamiltonian matrix and using initial guess methods like minao and 1e. When the Hamiltonian matrix is accurately predicted, the Self-Consistent Field (SCF) algorithm approaches convergence, resulting in a substantial reduction in the number of optimization steps. As a comparison, the optimal ratio calculates the single optimization step for each molecule divided by the total number of steps, serving as an illustrative benchmark of the ideal performance. Meanwhile, the error-level ratio calculates the ratio over the

Table 2: (Results updated by using the newest QHNet model.) The overall performance on the testing set on the defined four tasks. The unit for the Hamiltonian $\mathbf{H}$ and eigenenergies $\epsilon$ is Hartree denoted by $E_h$.

| Dataset | Model | $\mathbf{H}$ $[10^{-6}E_h]\downarrow$ | | | $\epsilon$ $[10^{-6}E_h]\downarrow$ | $\psi$ $[10^{-2}]\uparrow$ |
| | | diagonal | non-diagonal | all | | |
| --- | --- | --- | --- | --- | --- | --- |
| QH9-stable-id | QHNet | 111.21 | 73.68 | 76.31 | 798.51 | 95.85 |
| QH9-stable-ood | QHNet | 111.72 | 69.88 | 72.11 | 644.17 | 93.68 |
| QH9-dynamic-geo | QHNet | 149.62 | 92.88 | 96.85 | 834.47 | 94.45 |
| QH9-dynamic-mol | QHNet | 416.99 | 153.68 | 173.92 | 9719.58 | 79.15 |

number of steps required to reach a the same error level as model prediction during the DFT SCF loop compared to the total number of steps in the DFT process.

## 4 Experiments

**Setup.** To assess how deep learning approaches perform on the proposed dataset, we conduct experiments on the four designed tasks, as described in Section 3.2. To be more specific, we evaluate the performance of QHNet [Yu et al., 2023], a recently proposed SE(3)-equivariant network specifically designed for efficient and accurate quantum Hamiltonian matrix prediction. QHNet is known for its effectiveness and efficiency in handling the task at hand, making it a suitable testing method for our benchmark evaluation. For quantitative evaluation, we use the metrics as introduced in Section 3.4. Our implementation is based on PyTorch [Paszke et al., 2019], PyG [Fey and Lenssen, 2019], and e3nn [Geiger et al., 2022]. We train models on either (1) a single 48GB Nvidia GeForce RTX A6000 GPU and Intel Xeon Silver 4214R CPU, or (2) a single Nvidia A100 GPU and Intel Xeon Gold 6258R CPU.

Following the model setup in QHNet, in all implemented models, we employ five node-wise interaction layers to aggregate messages from neighboring nodes to update the node irreducible representations. We train all models with a total training step of either $210,000$ or $260,000$ using a batch size of 32. To expedite the convergence of model training, following the QHNet setup, we implement a learning rate scheduler. The scheduler gradually increases the learning rate from 0 to a maximum value of $5 \times 10^{-4}$ over the first $1,000$ warm-up steps. Subsequently, the scheduler linearly reduces the learning rate, ensuring it reaches $1 \times 10^{-7}$ by the final step.

**Overall performance.** We first evaluate the overall performance of the model on the four defined tasks by demonstrating its accuracy of the predicted Hamiltonian matrices on the testing set. As summarized in Table 2, the employed QHNet models can achieve a reasonably low MAE in predicting the Hamiltonian matrices on all proposed tasks. For reference, QHNet can achieve an MAE of $83.12 \times 10^{-6}E_h$ on the mixed MD17 dataset, which has a similar setup to our QH9-dynamic-geo setup. In addition to MAE on Hamiltonian matrices, the trained models also achieve low errors on the predicted occupied orbital energies and orbital coefficients. This aligns with the prior reported work that QHNet is effective to predict the Hamiltonian matrices for multiple molecules [Yu et al., 2023]. Notably, compared to the existing Hamiltonian matrix datasets, such as MD17 [Chmiela et al., 2017] and mixed MD17 [Yu et al., 2023], our proposed tasks involve predicting Hamiltonian matrices for significantly more molecules. Overall, we anticipate that the proposed new datasets and corresponding tasks can serve as more challenging and realistic testbeds for future research in Hamiltonian matrix prediction.

**Investigation on out-of-distribution generalization.** Since we maintain a similar number of training samples for QH9-stable-id and QH9-stable-ood, it is feasible to compare the performance of these two settings to investigate the out-of-distribution challenge in predicting Hamiltonian matrices. It is worth noting that we cannot directly compare the performance on their respective test sets, as reported in Table 2, to demonstrate the out-of-distribution generalizability challenge. This is because the molecules in the QH9-stable-ood test set have a larger number of atoms on average than those in QH9-stable-id. As explained in Section 3.4, molecules with larger size typically have more distant atom pairs, thus leading to a lower overall mean value of the Hamiltonian matrix. Hence, numerical results on molecules with different sizes are not directly comparable.

Table 3: The performance of in-distribution (ID) training and out-of-distribution (OOD) training on the constructed evaluation set for the OOD investigation.

| Training schema | Models | $\mathbf{H}$ $[10^{-6}E_h]\downarrow$ | | | $\epsilon$ $[10^{-6}E_h]\downarrow$ | $\psi$ $[10^{-2}]\uparrow$ |
| | | diagonal | non-diagonal | all | | |
| --- | --- | --- | --- | --- | --- | --- |
| ID | QHNet | 84.19 | 56.01 | 57.53 | 442.78 | 95.26 |
| OOD | QHNet | 113.05 | 70.52 | 72.78 | 630.49 | 94.01 |

Table 4: The performance of DFT calculation acceleration. Both models, trained on the QH9-stable-id split and the QH9-stable-ood split respectively, are evaluated on a common set of 50 randomly chosen molecules from the intersection of their test sets, making their results directly comparable. Similarly, results from models trained on the QH9-dynamic-geo and QH9-dynamic-mol splits can be compared directly.

| Training Dataset | DFT initialization | Metric | Ratio | Training Dataset | DFT initialization | Metric | Ratio |
| --- | --- | --- | --- | --- | --- | --- | --- |
| QH9-stable-id | 1e | Optimal ratio | $0.057_{\pm0.004}$ | QH9-stable-ood | 1e | Optimal ratio | $0.057_{\pm0.004}$ |
| | | Achieved ratio ↓ | $\mathbf{0.395}_{\pm0.030}$ | | | Achieved ratio ↓ | $0.400_{\pm0.030}$ |
| | | Error-level ratio ↑ | $\mathbf{0.635}_{\pm0.039}$ | | | Error-level ratio ↑ | $0.620_{\pm0.037}$ |
| | minao | Optimal ratio | $0.102_{\pm0.005}$ | | minao | Optimal ratio | $0.102_{\pm0.005}$ |
| | | Achieved ratio ↓ | $\mathbf{0.706}_{\pm0.031}$ | | | Achieved ratio ↓ | $0.715_{\pm0.033}$ |
| | | Error-level ratio ↑ | $\mathbf{0.408}_{\pm0.025}$ | | | Error-level ratio ↑ | $0.406_{\pm0.021}$ |
| QH9-dynamic-geo | 1e | Optimal ratio | $0.056_{\pm0.006}$ | QH9-dynamic-mol | 1e | Optimal ratio | $0.056_{\pm0.006}$ |
| | | Achieved ratio ↓ | $\mathbf{0.392}_{\pm0.036}$ | | | Achieved ratio ↓ | $0.512_{\pm0.138}$ |
| | | Error-level ratio ↑ | $\mathbf{0.648}_{\pm0.041}$ | | | Error-level ratio ↑ | $0.622_{\pm0.048}$ |
| | minao | Optimal ratio | $0.098_{\pm0.008}$ | | minao | Optimal ratio | $0.098_{\pm0.008}$ |
| | | Achieved ratio ↓ | $\mathbf{0.679}_{\pm0.041}$ | | | Achieved ratio ↓ | $0.882_{\pm0.217}$ |
| | | Error-level ratio ↑ | $\mathbf{0.443}_{\pm0.044}$ | | | Error-level ratio ↑ | $0.406_{\pm0.066}$ |

To examine the presence of the out-of-distribution issue in the Hamiltonian prediction task, we adopt an alternative evaluation strategy. To be specific, we assess models that have been trained respectively on the QH9-stable-id and QH9-stable-ood training sets, employing the same set of samples for evaluation in each instance. Specifically, we use the intersecting set of the QH9-stable-id and QH9-stable-ood testing sets as our evaluation dataset. Clearly, the samples contained within this evaluation set are previously unseen during the training phase of either model, thereby maintaining the integrity of the assessment. The evaluation set contains 923 molecules with 23 to 29 atoms. Under this experimental setup, the primary challenge faced by the model trained on the QH9-stable-ood training set stems from the novelty of molecular sizes during the evaluation phase. On the other hand, the model trained on the QH9-stable-id training set benefits from having been exposed to such molecular sizes during training. We denote that these two models are trained under out-of-distribution (OOD) and in-distribution (ID) training schema respectively in Table 3. By comparing the performance on the identical evaluation set, it becomes apparent that the model employing the ID training schema outperforms its OOD-trained counterpart, across all metrics including Hamiltonian MAE and predicted orbital energies and coefficients. Such a performance gap demonstrates that the out-of-distribution issue in molecular size is actually a valid concern particularly when extending trained models to molecular sizes not encountered during training.

**Geometry-wise *vs*. molecule-wise generalization.** We further explore geometry-wise and molecule-wise generalizability by analyzing the difficulty differences between the QH9-dynamic-geo and QH9-dynamic-mol tasks. We consider the results in Table 2 for these two tasks to be comparable given that both models are trained with a similar number of geometry structures. We note that the model in the QH9-dynamic-geo task demonstrates numerically better test performance than the model in the QH9-dynamic-mol task. This is consistent with our intention when designing the tasks. Specifically, in QH9-dynamic-geo, although the geometric structures in the test set are different, the molecules themselves are not entirely novel to the model due to the exposure during the training phase. In comparison, the QH9-dynamic-mol task presents a more challenging and demanding scenario. In particular, the test set geometries in QH9-dynamic-mol correspond to entirely different molecules than those seen during training. This task requires the model to generalize from its learned patterns to the unseen molecular structures. To summarize, both tasks serve as valuable testbeds in evaluating the model's generalization ability, and our analysis shows that QH9-dynamic-mol task, which requires extrapolating to entirely new molecular structures, is notably more challenging and demanding.

**Accelerating the DFT calculation.** We further measure the quality of the predicted Hamiltonian matrix by evaluating its ability in accelerating the DFT calculation. As introduced in Section 3.4, we

compute the ratio of optimization steps required when initializing with the predicted Hamiltonian matrix as compared to classic initial guess methods such as minao and 1e. Note that minao diagonalizes the Fock matrix obtained from a minimal basis to get the guess orbitals, and 1e is one-electron guess which diagonalizes the core Hamiltonian to obtain the guess orbitals. In this experiment, following our data collection process, we use PySCF [Sun et al., 2018] to perform the DFT calculation with using B3LYP exchange-correlation functional and def2SVP basis set. We select DIIS as the SCF algorithm for the DFT calculation and set a grid density level of 3 to ensure an accurate DFT calculation. For each dataset, we compute the average optimization step ratio for 50 randomly selected molecules. As shown in Table 4, we provide several metrics to reflect the optimization ratio. The optimal ratio is the ratio of a single step to the DFT optimization steps. The achieve ratio is the number of optimization steps when initialized by model prediction to optimization steps by DFT initialization. The error-level ratio is the number of optimization steps that achieve similar MAE error with model prediction to the total DFT optimization steps. We can observe that when initializing from the predicted Hamiltonian matrices given by QHNet, it requires fewer optimization steps to reach the converged Hamiltonian matrix, which indicates that the predicted Hamiltonian matrix is close to the convergence condition. This set of experimental results demonstrates that machine learning approaches are helpful in accelerating the DFT calculation.

## 5   Conclusion

We are interested in accelerating computation of quantum Hamiltonian matrices, which fundamentally determine the quantum states of physical systems and chemical properties. While various invariant and equivariant deep learning methods have been developed recently, current quantum Hamiltonian datasets consist of Hamiltonian matrices of molecular dynamic trajectories for only a single and four molecules, respectively. To significantly expand the size and variety of such datasets, we generate a much larger dataset based on the QM9 molecules. Our dataset provides precise Hamiltonian matrices for 130,831 stable molecular geometries and 999 molecular dynamics trajectories with sampled 100 geometries in each trajectory. Extensive and carefully designed experiments are conducted to demonstrate the quality of our generated data.

## Acknowledgements

This work was supported in part by National Science Foundation grants IIS-2006861, CCF-1553281, DMR-2119103, DMR-2103842, CMMI-2226908, and IIS-2212419 and by the donors of ACS Petroleum Research Fund under Grant 65502-ND10. Portions of this research were conducted with the advanced computing resources provided by Texas A&M High Performance Research Computing.

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

# A   Appendix

Table 5: Statistics of the datasets in Training/Validatioin/Testing datasets.

| Statistic Type | | QH9-stable-id | QH9-stable-ood | QH9-dynamic-geo | QH9-dynamic-mol |
|---|---|---|---|---|---|
| # of atoms | Mean | 18.02 / 18.02 / 18.03 | 16.97 / 21.25 / 23.67 | 16.53 / 16.53 / 16.53 | 16.56 / 16.38 / 16.38 |
| | Median | 18 / 18 / 18 | 17 / 21 / 23 | 17 / 17 / 17 | 17 / 17 / 17 |
| | Max | 29 / 29 / 29 | 20 / 22 / 29 | 19 / 19 / 19 | 19 / 19 / 19 |
| | Min | 3 / 6 / 4 | 3 / 21 / 23 | 10 / 10 / 10 | 10 / 10 / 10 |
| # of electronics | Mean | 65.89 / 65.90 / 65.86 | 64.98 / 68.83 / 70.57 | 64.68 / 64.68 / 64.68 | 64.71 / 64.44 / 64.68 |
| | Median | 66 / 66 / 66 | 66 / 70 / 70 | 66 / 66 / 66 | 66 / 66 / 66 |
| | Max | 74 / 74 / 74 | 74 / 74 / 74 | 74 / 74 / 74 | 74 / 70 / 70 |
| | Min | 10 / 18 / 24 | 10 / 56 / 58 | 38 / 38 / 38 | 40 / 38 / 46 |
| Size of Hamiltonian | Mean | 141.59 / 141.62 / 141.56 | 138.99 / 149.85 / 155.09 | 137.96 / 137.96 / 137.96 | 138.04 / 137.34 / 137.88 |
| | Median | 144 / 144 / 144 | 142 / 150 / 154 | 140 / 140 / 140 | 142 / 142 / 142 |
| | Max | 166 / 166 / 166 | 148 / 152 / 166 | 146 / 146 /146 | 146 / 146 / 146 |
| | Min | 18 / 36 / 48 | 18 / 126 / 130 | 82 / 82 / 82 | 86 / 82 / 92 |

Table 6: Test performance for the QHNet trained on QH9-stable-id and QH9-stable-ood datasets on the intersection of test sets over various molecule size.

| # of Atoms | # of Samples | Training schema | $\mathbf{H}$ $[10^{-6}E_h]$ ↓ | | | $\epsilon$ $[10^{-6}E_h]$ ↓ | $\psi$ $[10^{-2}]$ ↑ |
|---|---|---|---|---|---|---|---|
| | | | diag | non-diag | all | | |
| 23 | 620 | ID | 86.22 | 57.75 | 59.31 | 435.88 | 94.83 |
| | | OOD | 107.09 | 69.87 | 71.92 | 583.76 | 93.47 |
| 24 | 82 | ID | 92.25 | 58.53 | 60.31 | 353.93 | 96.92 |
| | | OOD | 125.20 | 75.29 | 77.93 | 365.96 | 96.74 |
| 25 | 180 | ID | 76.73 | 50.99 | 52.30 | 505.51 | 96.34 |
| | | OOD | 118.84 | 69.56 | 72.07 | 1135.47 | 94.38 |
| 26 | 4 | ID | 79.75 | 51.49 | 52.88 | 254.94 | 97.66 |
| | | OOD | 131.86 | 72.89 | 75.79 | 406.69 | 95.73 |
| 27 | 33 | ID | 69.33 | 46.10 | 47.21 | 264.35 | 95.27 |
| | | OOD | 146.63 | 73.43 | 76.90 | 491.40 | 91.97 |
| 28 | 0 | ID | – | – | – | – | – |
| | | OOD | – | – | – | – | – |
| 29 | 4 | ID | 71.42 | 45.87 | 47.00 | 243.20 | 94.55 |
| | | OOD | 223.41 | 90.25 | 96.15 | 772.10 | 90.61 |

Table 7: Test performance with various training set sizes.

| Training sets | # of examples | Models | $\mathbf{H}$ $[10^{-6}E_h]$ ↓ | | | $\epsilon$ $[10^{-6}E_h]$ ↓ | $\psi$ $[10^{-2}]$ ↑ |
|---|---|---|---|---|---|---|---|
| | | | diag | non-diag | all | | |
| QH9-Stable-id | 104, 664 | QHNet | 111.21 | 73.68 | 76.31 | 798.51 | 95.85 |
| QH9-Stable-id-50k | 50,000 | QHNet | 116.23 | 74.26 | 77.20 | 1127.33 | 95.37 |
| QH9-Stable-id-10k | 10,000 | QHNet | 127.07 | 74.93 | 78.60 | 2471.46 | 96.18 |

## A.1   Detailed dataset statistics

Here we provide a detailed statistics including the number of atoms, the number of electronics and the size of Hamiltonian matrix for training, validation and test sets in Table 5 for all the proposed datasets.

## A.2   Test performance over molecule size

To evaluate the model performance and alleviate the influence of molecular size, we test the QHNet trained on QH9-stable-id and QH9-stable-ood datasets, and demonstrate the test performance on the molecule with various sizes in Table 6.

## A.3 Test performance over various training sets

To investigate the model performance with various number of examples in training sets, we train the QHNet on datasets with various sizes. As shown in Table 7, we can observe that while the MAE on Hamiltonian matrix and cosine similarity on wavefunction $\psi$ are in similar level with various training sets, the MAE on occupied eigen energies $\epsilon$ becomes much worse when the training sets become smaller.

