# OpenReview forum: "QH9: A Quantum Hamiltonian Prediction Benchmark for QM9 Molecules"
_NeurIPS.cc/2023/Track/Datasets_and_Benchmarks — NeurIPS 2023 Datasets and Benchmarks Poster_

### Official Review · Reviewer_gi6S · 2023-07-20
**Significant advance, some room for improvement**

**Rating:** 6
**Confidence:** 4

**Strengths:**

The main strenght of this submission is providing a comprehensive dataset of Hamiltonian matrices that allows assessing the reliability of machine learning modles for the prediction of Hamiltonian matrices in a more realistic way. This allows assessing the state of the art in the field in a more representative way and is a stepping stone towards more accurate models to be developed in the future. Additionally, the source code is provided in a readily usable way allowing simple use of the provided dataset for training alternative models and testing their prediction accuracies.

**Additional Feedback:**

There is no additional feedback.

**Clarity:**

The paper is well written and generally clear. However, there is one sentence in particular that I did not understand. The authors state the following: "We implement the GTO basis set Def2SVP [Weigend and Ahlrichs, 2005] in a post-HF method that incorporates aspects of DFT, namely, an exchange-correlation potential, in order to more accurately capture electron-electron interactions compared to the HF method, which uses a mean field approximation of electron density." What post-HF method do the authors refer to? Do the authors refer here to B3LYP? If so, I think the authors should simply state that to avoid any confusion.

**Correctness:**

I think there are a few minor issues:
1. Unlike stated in the Background and Related Works session, I do not think that the HOMO-LUMO gap is a physical property but rather a virtual property.
2. The authors name the HeN term of the Hamiltonian the electron-ion interactions, I think electron-nucleus interactions is a more appropriate term.

**Documentation:**

The necessary details regarding the dataset are provided in sufficient detail. The GitHub repository provides clear documentation for usage of the dataset and a license. The code itself could benefit from better commenting.

**Ethics:**

I do not have any ethical concerns with the submission.

**Limitations:**

As the authors also indicated in their checklist, they do not describe the limitations of their work.

**Opportunities For Improvement:**

1. In the introduction, the authors state correctly that pure DFT scales as O(N3). However, in the manuscript the authors make use of B3LYP, which incorporates both HF and DFT and thus formally probably rather scales as O(N4). While it is implicitly mentioned in the text, I think it would be good for the reader to point this out explicitly.
2. I think it would be insightful to point out how the molecules contained in QH9 were selected.
3. I think it would be insightful to vary the training set size and generate learning curves. This will allow assessing what training dataset size would be required for the model to reach the desired prediction accuracies.
4. For deriving the optimization step ratio benchmark metric, the authors claim that "it calculates the ratio of the number of optimization steps between initializing with the predicted Hamiltonian matrix and using a random initialization." However, based on the code provided in the GitHub repository, the baseline is not a random initialization but an initialization based on a guess in terms of the overlap to minimal basis ('minao') as this is the default setting when no guess matrix is provided.
5. When comparing the opimization step ratio, I think it would be insightful to not only provide the average but also the standard deviation across the selected molecules.
6. I think it would be important to add additional baselines to the tables. For instance, for the optimization step ratio, I think the lowest achievable optimization step ratio, which would essentially be 1 step for each of the molecules divided by the total number of steps, should also be provided in this table. As the number of optimization steps tends to depend on the molecule size, it depends on the molecules selected and thus this lowest achievable step ratio is important for providing context. Additionally, classical baselines should also be provided. One could think of performing a single point calculation with a minimal basis and using the corresponding result to derive a guess matrix. Furthermore, alternative ways to compute a guess matrix are possible and I think it would be useful to provide them as well. For tables 2 and 3, I think it would be good for the reader to add a row that provides the accuracy that would correspond to SCF convergence which would be the desired target accuracy. Additionally, I think it would also be useful to provide the accuracy of guess matrices as calculated by standard methodology such as 'minao' as well as these entries will provide context for both upper and lower boundaries.

**Relation To Prior Work:**

I think the differences to prior work are clearly worked out and the advances provided by this submission are emphasized.

**Summary And Contributions:**

The article "QH9: A Quantum Hamiltonian Prediction Benchmark for QM9 Molecules" describes the generation of a comprehensive dataset of Hamiltonian matrices for a large number of molecules and a considerable number of molecular dynamics trajectories. Additionally, this dataset is demonstrated for training machine learning models that are subsequently evaluated with respect to their prediction accuracy.

---

> ### Author Response · Authors · 2023-08-25
> **Thanks for your comments**
>
> 1. Thanks for pointing it out. I have added descriptions about the complexity using B3LYP in our revised version to clarify the complexity and avoid confusion. Thanks for the constructive suggestions.
> 2. The molecule of the QH9 comes from QM9 datasets. For the QH9-static, we generate the corresponding Hamiltonian matrix for all the molecules from QM9. For the QH9-static, we randomly select the molecules with no more than 20 atoms to generate the molecular dynamics.
> 3. Thanks for your comments. We provide the results of selecting 10k, 50k and all molecules from the training sets of QH9-static-iid. From the table, we can observe that while the Hamiltonian matrix and $\psi$ are in similar level, the $\epsilon$ is much worse when the training sets become small.
> 	|  Dataset |  diag | non-diag | all | $\epsilon$ | $\psi$ |
> 	| -------- | ------- |  -------- | ------- | -------- | ------- |
> 	| QH-Stable-iid 	| 111.21	| 73.68	| 76.31	| 798.51 | 95.85 |
> 	| QH-Stable-iid-50k 	| 116.23          	| 74.26 | 77.20 | 1127.33 | 95.37 |
> 	| QH-Stable-iid-10k 	| 127.07	| 74.93 | 78.60 | 2471.46 | 96.18 |
> 4. Thanks for pointing it out. For the original baseline, we wish to claim that we provide the optimization step ratio between using the prediction of quantum tensor networks and the ‘minao’ random guessing initialization.
> 5. Thanks for your comment. We add the standard deviation in Table 4.
> 6. Many thanks for the constructive comments. We provide a comprehensive comparison on the optimization ratio. First, we provide the lowest achievable ratio. Then, we wish to clarify that ‘minao’ random guessing initialization actually takes use of atomic densities projected in a minimal basis. But we do think providing more classic baselines is important. Therefore, we provide the optimization ratio of ‘1e’ random guessing initialization which takes the diagonalization of core Hamiltonian to do initialization. Meanwhile, we provide the  error-level ratio which is the optimization ratio when the SCF algorithm obtains the Hamiltonian matrix with a similar error level compared to the Hamiltonian matrix predicted from the QHNet.
>
> Minor issue 1: Thanks for your opinion, and we have revised and deleted the HOMO-LUMO gap.
> Minor issue 2: Thanks for pointing it out. We have changed the description and notation based on your advice.
> Clarity: We wish to clarify that post-HF is another category of methods that further improve the performance of HF by techniques like configuration interaction, perturbation and so on. For better understanding, we delete these words in our revised manuscript to avoid confusion.

---

> > ### Comment · Reviewer_gi6S · 2023-08-25
> > **Thanks for your response**
> >
> > Thank you for addressing most of my comments. Here are still a few things that I thing need changing in the revised paper:
> >
> > 2. I could not find it explicitly mentioned in the text that QH9 encompasses all the molecules of QM9. Could you still add this to make it clear to the reader?
> > 3. Thank you for these results. I think they are quite insightful. Could you add them to the manuscript, at least to the Supplementary Material? I could not find it there in the updated version.
> > Minor issue 1: As far as I can see, you removed the HOMO-LUMO gap but you now claim that HOMO and LUMO are physical properties. I think they are rather virtual properties as they cannot be observed experimentally.
> >
> > Once these issues are properly addressed, I will raise my score.

---

> > > ### Author Response · Authors · 2023-08-25
> > > **Thanks for posting your further comments**
> > >
> > > 2. Thanks for the comment. We have modified the description in section 3.1 to describe the source of molecules and their geometries for  QH-stable dataset. We provide the content as below.
> > > * We obtain the molecules and their geometries in QH-stable from the QM9 subset with 131,831 molecules which is widely used in molecular property prediction tasks of previous studies including Schütt et al. [2018], Gasteiger et al. [2020], Liu et al. [2022], Wang et al. [2022]. Note that we make use of all the molecules in this QM9 subset.
> > > 3. Thanks for this advice. We have added this experiment in Table 7 in appendix.
> > >
> > > Minor issue 1: We wish to clarify that HOMO from DFT calculation is directly related to the ionization energy (which is experimentally measurable), according to Jamal's theorem [1]. For the LUMO, it does not correspond to exact physical property like HOMO, but since it is approximately related to electron affinity, we provide the example of LUMO here.
> > >
> > > [1] Janak, James F. "Proof that∂ E∂ n i= ε in density-functional theory." Physical Review B 18.12 (1978): 7165.

---

> > > > ### Comment · Reviewer_gi6S · 2023-08-26
> > > > **Thank you for the clarifications**
> > > >
> > > > Thank you for addressing all my comments. Consequently, I raised my review score.

---

### Official Review · Reviewer_wfG7 · 2023-07-24
**Excellent contribution to computational chemistry**

**Rating:** 6
**Confidence:** 3

**Strengths:**

I think the authors have assembled a very relevant dataset and their observation -- fundamentally, that there are novel demands on network design owing to the symmetry requirements in Hamiltonians -- is a valid one, so the dataset and benchmark will remain relevant for some time.

**Additional Feedback:**

No additional feedback beyond the above. Thank you for the read!

**Clarity:**

The paper is generally well written, but I am concerned that some of my opportunities for improvement stem from misunderstandings that may arise from the paper's text itself.

**Correctness:**

I believe, with moderate confidence, that the benchmark was conducted appropriately. I am reasonably confident that the dataset is constructed soundly but it could have been constructed better.

**Documentation:**

I believe the maintenance plan is implicit (it's on Google Drive) but I don't know if that link's going to work forever, and there is no explicit plan for how it will remain available for the future.

**Ethics:**

No such suspicion.

**Limitations:**

In re "Have the authors adequately addressed the limitations and potential negative societal impact of their work?": the author checklist responds "no"; I generally agree with them. I believe the paper would have been strengthened by explicit discussion of these limitations and the possibility of further work to eliminate them.

**Opportunities For Improvement:**

I believe that the particular instantiation of this dataset could have been constructed better, and so the abstraction of "a dataset on small molecule Hamiltonians" is carrying the day better than the specific data that is available right at this moment, or its presentation in the manuscript. Some open questions:
* Why were a fraction of QM9 molecules excluded from calculations? 134k molecules became <131k, but the reasons for excluding 3k (while possibly trivial) are not given.
* An ood split based on number of total atoms (as in QH-stable-ood) is reasonable, but many molecules may be highly similar across the split boundary. It would be valuable to see the Tanimoto similarity (for example) across split boundaries, and (if many validation examples have highly similar training examples) to evaluate performance on splits designed to reduce maximum Tanimoto similarity for validation examples.
* The cursory description of the MD data generation process leaves something to be desired. NVE dynamics are conducted, but the force field is not specified. Is this ab initio MD or does it use a classical force field and the individual geometries are then re-optimized in DFT? How structurally diverse (e.g,, in heavyatom RMSD) are the sixty resulting frames; how long were the trajectories run, and are the frames time-correlated? Why was such an unusual time step chosen (2.4 attoseconds); even for AIMD I've seen 0.1 fs quite commonly.

**Relation To Prior Work:**

Yes, the authors clearly give credit to prior art and differentiate themselves well.

**Summary And Contributions:**

Authors take the vast majority of molecules in QM9 -- a pre-existing benchmark of molecular geometry, including all organic compounds with nine or fewer heavyatoms (comprising elements CHONF) -- and compute Hamiltonian matrices. Authors further perform MD simulations on 2399 molecules in microcanonical ensemble for another significant contribution. Authors supply iid and ood splits for future use, provide well-justified metrics for evaluation, and perform baseline experiments with QHNet.

---

> ### Author Response · Authors · 2023-08-25
> **Thanks for your comments**
>
> 1. Thanks for pointing it out. The original QM9 dataset contains 134K molecules, but we use a subset of QM9 with 13,831 molecules which is widely used to evaluate the energy prediction models such as SchNet, Dimnet, SphereNet and so on. Instead of using the original split of 110,000/10,000/10,831, we take a random split for the QH-stable-iid to keep the ratio of examples in training/validation/test sets be 0.8/0.1/0.1. We wish to clarify that there is no strong reason about which dataset split is better, and we use random split for convenience.
> 2. Thanks for your comments. It is a very interesting idea to apply Tanimoto similarity. We first wish to clarify that we obtain our QH-stable-ood datasets based on the molecule sizes. And we have evaluated that such OOD may cause the model performance to drop as shown in Table 3. Then, we provide the similarity here for better evaluation based on your suggestion. We provide the average Tanimoto similarity calculated from all the example pairs. In QH-stable-ood, the atom numbers of molecules in training sets are no more than 20 and the atom numbers of molecules in testing sets are no less than 23. Therefore, we provide the Tanimoto similarity between the molecule pairs with size 20 and 23. We can observe that the  Tanimoto similarity is less compared to the molecules with the same size.
> |  Description	| Tanimoto similarity |
> | -------- | ------- |
> | Between 20 and 23 | 0.3656 |
> | Between 20 and 20 | 0.4050 |
> | Between 23 and 23 | 0.4320 |
>
> 3. Thanks for your comments. First, we wish to clarify that MD data is generated by ab initio molecular dynamics calculations using the DFT method implemented in PySCF.  That means, all the energies and atomic forces are directly calculated by DFT using the Hellmann-Feynman theorem. No classical force fields were used. With a 2.4 attosecond as time step, the total trajectory is 0.144fs and the frames are time-correlated. These datasets consider short-time vibrations around the molecular equilibrium structure. With your advice, we are currently conducting MD simulations with longer time steps (0.12fs) and longer trajectories (1000 steps), and we plan to release this dataset that represents longer molecular trajectories when it is finished.
>
> Documentation: We understand your concern. We plan to release our dataset on zenodo with DOI for camera ready version and we have already reserve the DOI 10.5281/zenodo.8274793 for the QH9 dataset.

---

### Official Review · Reviewer_jeVm · 2023-07-24
**Quantum Hamiltonian Dataset**

**Rating:** 7
**Confidence:** 4

**Strengths:**

As mentioned above:
1. a reference dataset for predicting the electronic energy of molecule system
2. using a known dataset MQ9
3. the description of the format and representation of the energy

**Additional Feedback:**

I hope I provided enough feedback.

your work is very welcome!

**Clarity:**

The paper is relative clear, the highlighted point is because more detail are necessary in the methematical description and the relationship to the data.

**Correctness:**

I was not able to run the dataset. The authors provide a link to the dataset repository, I feel the documentation on the installation could be improved.

**Documentation:**

The authors need to give installation instruction also because this is required to download the dataset.

Script to generate the data (extend) would be better if provided.

The dataset need to be uploaded in a permanent DOI, currently is only available in google drive, which can not easily installed in servers.

**Ethics:**

No concerns

**Limitations:**

The authors did not discuss the negative impact, but a part from providing not correct data, I do not see direct negative impact. of course the use of the train model can be used for many applications.

**Opportunities For Improvement:**

I feel the following improvement needs to be implemented:
1. Extend the presentation session on the mathematical representation for the Quantum Hamiltonian. I thin the current explanation is not complete. Is the represenation unique? which other (main) representation exist and how to convert among them if possible and if not why.
2. in equation (1) ri is probably the coordinate of the electron, please give more details, why is not the atomic information also included and probably you can mention but drop the reference. the ri coordinates probably only make sense in relation to the R coordinates of the main atoms.
3. you mentioned few properties, may be you can provided reference and intuition on how these are derived from the wave function or from the H described later
4. In general is not clear what is provided in the dataset of the variables described in (2) and (3).
5. in section 2.2 you introduce the equivariance, it is not clear what is "\ell". Please provide reference to equation (4), a description of the properties of the order -\ell Wigner-D matrix and a reference.
6. in line 126 you describe the H. Is this specific of the SCP or is general? this is not clear; further you talk about \ell_j as angular momentum but then use as index in the block matrix, this does not seem to be correct, could you please give more information?
7. please give more detail about Table.1 (for example: mean, min, max, median number of atom, electron, size of the H matrix; Computation time.the size of the systems seem small.)
8. why only 9 (heavy) atoms?
9. Could you explain why you did not use the FermiNET and the recent models?
10. why in table 2 OOD has better performance of IID? (line 2, last columns)
11. In general you describe the convergence criteria, but how did you validate that the results are correct?
12. molecular dynamic with 60 samples it is very small. How do you justify (a part the numerical complexity? and how do you validate that this samples are representative, the time step seems quite small
13. please provide installation instruction (possibly environment and pip local installation)
14. please add the dataset to a permanent (DOI) url
15. please provide instructions on how to extend the dataset
16. please provide the script on how to run the SCP computations (and data transformation)

Even if the list is long, I am looking for the paper will be accepted to the NeurIPS track.

**Relation To Prior Work:**

Yes, it does.

**Summary And Contributions:**

The paper presents a dataset of Quantum Hamiltonian Prediction dataset for set of molecules from a well known public dataset (QM9).

The dataset provided contains two set of data: the structure with energy and the molecular dynamic simulation data.

The difference with existing available dataset is that QH9 provide information about the electric state that is typically computed using Quantum Mechanical solvers.

The structure of the representation is thus different from the standard molecular representation, since the energy and configuration of the electron need to be provided.

This dataset is a relevant step towards a better description and prediction of molecular states.

The authors provide a short mathematical background of the representation used and the description of the solution method and the dataset itself.

The dataset thus will help the development and validation of new ML model to predict the energy of the electronic state of molecules.

---

> ### Author Response · Authors · 2023-08-25
> **Thanks for your feedback - reply 1**
>
> Opportunities For Improvement:
> 1. If we understand correctly, we provide more details about Equation (3) to explain the Hamiltonian matrix. Equation (3) contains electron-electron Coulomb interaction, electron-ion interaction, and the exchange-correlation energy. Without external electric and magnetic fields, Equation (3) is the complete electronic Hamiltonian that determines the underlying electronic structure of materials. To numerically represent Hamiltonian in a matrix, one needs to choose a specific basis set. It can be real-space grids, plane-wave basis, or localized basis. We choose the “def2SVP '' Gaussian Type Orbital (GTO) as the local basis to calculate the Hamiltonian matrix, and the eigenvectors correspond to the wave function coefficients. One can convert the Hamiltonian matrix approximately from the GTO basis to the STO or plane-wave basis. And we will provide more mathematics if the conversion details are needed. For each entry in Hamiltonian matrix, $H_{ij} = < \psi_i | \hat{H} | \psi_j > $. Note that in Equation (5), the $\mathbf{H}_{ij}$ denotes a matrix block, while $H_{ij}$ denotes an element.
> 2. We wish to clarify that $\Psi$ denotes the electronic wavefunction. Therefore, the input of this wavefunction is the coordinates of the electrons. By the way, the atomic information is used when calculating the Hamiltonian matrix. In the Equation (3), the electron-electron interactions $\mathbf{H}_{eN}$ encode the interaction between the atoms and the electrons.
> 3. Here we provide several examples. For the energy band, when performing the eigen decomposition shown in Equation (2), the corresponding $\epsilon$ represents the energy band. For the HOMO energy, it is the energy that corresponds to the highest occupied in these energy bands and LUMO is the energy that corresponds to the lowest unoccupied in these energy bands.
> 4. We wish to clarify that we provide the Hamiltonian matrix $\mathbf{H}$ in our datasets. We have added descriptions in Section 2.1 to provide more details.
> 5.
> * We wish to clarify that $\ell$ denotes the rotation order. Mathematically, the SO(3) equivariant features have dimensions $2\ell + 1$ for $\ell \in \mathbb{N}$ (including $\ell = 0$) and are unitary. Physically, $\ell$ corresponds to the angular quantum number of the molecular orbitals. We introduce their relationship in the next question. This definition is provided in [1].
> * Equation (4) is obtained from the standard definition of group equivariance. Actually, for any group $g$, many literatures [2,3,4] describes the equivariance of a function $f$ w.r.t. $g$ as $f(S(x))=T(f(x))$, where $S, T$ are transformations of $g$. For the SE(3) group, the standard equivariance is to use 3D rotation and translation for $S$ and rotation by Wigner-D matrix for $T$, which results in Equation (4).
> * Order-$\ell$ Wigner-D matrix is an irreducible representation of rotation group SO(3), and represents the rotational equivariance of commonly used spherical harmonics $Y^\ell$ as $D^\ell(R)Y^\ell(x)=Y^\ell(Rx)$. See Section A.3 of [3] and Section A.2.1 of [4] for more background information.
> - [1] Tensor field networks: Rotation- and translation-equivariant neural networks for 3D point clouds. arXiV: 1802.08219.
> - [2]  SE(3)-Transformers: 3D Roto-Translation Equivariant Attention Networks. NeurIPS 2020.
> - [3] Geometric and Physical Quantities Improve E(3) Equivariant Message Passing. ICLR 2022.
> - [4] Equivalence Between SE(3) Equivariant Networks via Steerable Kernels and Group Convolution. arXiV: 2211.15903.
> 6. We are willing to provide more information. First, $\mathbf{H}$ in line 126 is the Hamiltonian matrix obtained from DFT calculation including SCF, and we add brief words in our revised version. Then, thanks for pointing out the confusion about angular momentum, and we revise it as angular quantum number. As a brief introduction, principal ($n$), angular ($\ell$), and magnetic ($m$) quantum numbers are three important physical properties to describe the orbital wave function. There is a relationship between the angular quantum number of the orbital and the rotation order of corresponding equivariant features or matrices. For example, considering the widely used spherical harmonics to encode the pairwise directions, the angular quantum number of the spherical harmonics is the same as the rotation order of obtained equivariant features. Therefore, we use the same notation to denote the angular quantum number and rotation order considering such a relationship. Moreover, the index in the block matrix is used to denote the rotation order of the Wigner-D matrix.

---

> > ### Author Response · Authors · 2023-08-25
> > **Thanks for your feedback - reply 2**
> >
> > 7. Thanks for your advice. We provide these statistics in table 5 in appendix. For the training time, our models were trained using the NVIDIA A100 GPU and the Intel Xeon Gold 6258R CPU clocked at 2.70GHz. On an idle CPU, training a model for 300,000 iterations with a batch size of 32 takes approximately 4 days.
> > 8. We wish to clarify that we use the molecules from QM9 which contain molecules with no more than 9 heavy atoms.
> > 9.
> > * We wish to point out that we discuss recent models on quantum tensor learning tasks in Section 3.3. For other baseline models, since QH9 contains various molecules in training and test sets, which is a new case for previous methods, some modifications are needed when applying them on QH9. For example, PhiSNet may need to change the blockwise Hamiltonian construction module and the channel indices dictionary to fit Hamiltonian matrices with various sizes. As our future plan, we will continue to develop our benchmark to consider these methods with modifications.
> > * We wish to clarify that FermiNET is not designed for predicting Hamiltonian matrix prediction which is kind of out of our scope currently.
> > 10. Thanks for pointing it out. We have updated our results based on the newest version of QHNet. We provide our new results in our revised submission and you can find the IID has better performance.
> > 11. Thanks for your question. When the convergence conduction is achieved, the corresponding algorithm will provide a bool flag to show it. We will check the convergence flags for all the DFT computations.
> > 12. Thanks for your comments. Currently, the provided MD trajectory is 0.144fs in total with 2.4 attosecond as time step which reflects short time vibrations around the molecular equilibrium structures. And we are currently running MD simulations with longer time steps (0.12fs) and longer trajectories (1000 steps), and we plan to release this dataset that represents longer molecular trajectories when it is finished.
> > 13. Thanks for your comments. Currently, we are a github repository that can be easily modified based on users’ needs. We provide install files to build the environment and packages in [install.sh](https://github.com/divelab/AIRS/blob/main/OpenDFT/QHBench/QH9/install.sh). We will plan to make it into a package in the futher.
> > 14. Thanks for your advice, we have reserved the DOI 10.5281/zenodo.8274793 for the QH9 dataset, and we will upload our datasets in camera ready version.
> > 15. Thanks for your advice. We provide comments on how to build your own datasets in our [github repository](https://github.com/divelab/AIRS/tree/main/OpenDFT/QHBench/QH9#customization). We sincerely hope it can help users build their own dataset.
> > 16. Thank you for your comments. I guess you mean the SCF computation. For this computation, we provide the scripts of calculating Optimization ratio which is used to perform SCF loop. The scripts are shown in [test_dft_acceleration](https://github.com/divelab/AIRS/blob/main/OpenDFT/QHBench/QH9/test_dft_acceleration.py).
> >
> > Correctness and Documentation: Thanks for giving the feedback. We provide the benchmark codes in our [github repository](https://github.com/divelab/AIRS/tree/main/OpenDFT/QHBench/QH9), and revise our installation instruction in Readme. Currently, we save our datasets in [Google Driver](https://drive.google.com/drive/folders/13pPgBh3XvN2FCpowfnA8TT4VJ0OTceNM). We plan to release our dataset with DOI for camera ready version and we have already reserve the DOI 10.5281/zenodo.8274793 for the QH9 dataset.

---

> > > ### Comment · Reviewer_jeVm · 2023-08-31
> > > **DOI and original script**
> > >
> > > Dear authors,
> > >
> > > thank you for the feedback and the action times. I am happy to increase my score. It is important the dataset to have the DOI and available not only via the google drive and the framework to be extensible as well clear installation instruction. Thank you for your effort also in clarifying the Hamiltonian part. I hope you reflected in the text the clarifications.
> > >
> > > Best Regards

---

### Official Review · Reviewer_R63V · 2023-07-28

**Rating:** 7
**Confidence:** 2

**Strengths:**

- The dataset seems useful, but I do not know enough about the problem to validate this exactly.

**Additional Feedback:**

I gave the score good paper because there is nothing to me that stand out as obviously bad from an ML perspective. My knowledge of quantum chemistry is however limited which makes it difficult for me to evaluate this.

**Clarity:**

Overall the writing is clear. I do believe that it is quite hard to follow all the physics concepts for a person with a non physics background.

**Correctness:**

The paper seems correct. My knowledge of the underlying physics problem is limited however.

**Documentation:**

Link to the data is missing in the main paper.

**Ethics:**

No issue.

**Limitations:**

No societal negative impact to be discussed.
For comments see opportunities for improvement.

**Opportunities For Improvement:**

- Only one model is evaluated and only on the proposed dataset. It would be helpful if it can be shown that the model obtains good results on previous datasets and that the results presented here are in line with what can be expected based on prior results or an explanation of why what is not the case.

- It is hard to me to understand which problem is the most difficult. From the discussion it seems that for larger molecules the metrics (energy) goes down on average anyway. Would it make more sense to evaluate the molecules per size independently or is this average metric still useful.

- In Table 4 it seems that for all problems the practical effect on speeding up DFT computation is quite similar regardless of the setup. Why were only 50 randomly selected molecules used and were it the same molecules in each setup or was there a difference between setups.


**Relation To Prior Work:**

I have not followed this

**Summary And Contributions:**

The contribution are:
- a new dataset of hamiltonian matrices for 2399 molecular dynamics trajectories and for 130.831 stable geometries.
- the evaluation of a machine learning model on this dataset.

---

> ### Author Response · Authors · 2023-08-25
> **Feedback to the review**
>
> Opportunities For Improvement:
> 1. Thanks for your comments. We understand that reasonable evaluation performance is important. For other baseline models, since QH9 contains various molecules in training and test sets, which is a new case for previous methods, some modifications are needed when applying them to QH9. For example, PhiSNet may need to change the blockwise Hamiltonian construction module and the channel indices dictionary to fit Hamiltonian matrices with various sizes. As our future plan, we will continue to develop our benchmark to consider these methods with modifications. Currently, we only provide QHNet as the baseline which is suitable for this dataset.
>
> 2. Thanks for your comments.
> * For the difficulty of proposed datasets, the QH-stable-ood task is more difficult than QH-stable-iid. From the experiment results in Table 3, you can find that test performance is much worse in the OOD settings with the same molecules in the test sizes. For the QH-dynamic-geo and QH-dynamic-mol, it is intuitive to think that QH-dynamic-geo is a little easier since the molecules in testing sets have been seen in training sets, while the molecules and geometries are all different in QH-dynamic-mol. However, since the test sets of these two are different, the test performance between these two datasets can not be used as a rigorous comparison to show the difficulty.
> * We wish to clarify that this average metric provides a way to compare different baselines and reflect the quality of the predicted Hamiltonian matrix to some extent. For example, in Table 2, these average metrics show that QHNet trained on QH-Stable-iid consistently performs better than QHNet trained on QH-Stable-ood.
> * Since testing performance per size can alleviate the influence of molecule size on the metrics, we further provide the test performance for each molecule size on the intersection test sets of QH-Stable-iid and QH-Stable-ood. The results are shown in Table 6 in appendix.  From this table, we can observe that QHNet trained on QH-Stable-iid consistently performs better than QHNet trained on QH-Stable-ood. Meanwhile, for the QHNet trained on QH-Stable-iid, there is an approximate tendency of smaller MAE on non-diagonal blocks in the Hamiltonian matrix, which demonstrates the influence of molecule size.
>
> 3.  We wish to clarify that when calculating the DFT acceleration ratio, the molecules are randomly selected from test sets. Meanwhile, for QH-dynamic-geo and QH-dynamic-mol, they share the same molecular geometries from the intersection of their test sets, and for QH-stable-iid and QH-stable-ood, they use the same examples from the intersection of their test sets. Considering the cost of the DFT algorithm, we only select 50 molecules when calculating the optimization ratio.
>
> Documentation:
> Thanks for pointing it out. We will add the data link and DOI in the main paper for camera ready version.

---

### Author Response · Authors · 2023-08-25
**Thanks for your advice**

Dear reviewers and ACs,

Thanks for your constructive advice. Based on the suggestions, we have revised our paper noted by red, updated our experiment results with the newest version of QHNet and provided detailed comments to make this work better. We are sorry for the late reply since we take lots of time to conduct the experiments, revise the paper and provide detailed feedback. Thanks for your consideration as well as your time and efforts.

Best,

Authors

---

### Decision · Program_Chairs · 2023-09-22

**Decision:**

Accept (Poster)

**Comment:**

The reviewers all appreciate the contributions of the paper.

The discussion lead to further clarification and improvement of the contribution.